# Testing the skill-based approach: Consolidation strategy impacts attentional blink performance

**Corné Hoekstra**[1]*, **Sander Martens**[2], **Niels A. Taatgen**[1]

**1** Bernoulli Institute for Mathematics, Computer Science, and Artificial Intelligence, University of Groningen, Groningen, The Netherlands, **2** Department of Biomedical Sciences of Cells and Systems, University Medical Center Groningen, University of Groningen, Groningen, The Netherlands

* c.hoekstra@rug.nl

## Abstract

Humans can learn simple new tasks very quickly. This ability suggests that people can reuse previously learned procedural knowledge when it applies to a new context. We have proposed a modeling approach based on this idea and used it to create a model of the attentional blink (AB). The main idea of the skill-based approach is that models are not created from scratch but, instead, built up from reusable pieces of procedural knowledge (skills). This approach not only provides an explanation for the fast learning of simple tasks but also shows much promise to improve certain aspects of cognitive modeling (e.g., robustness and generalizability). We performed two experiments, in order to collect empirical support for the model's prediction that the AB will disappear when the two targets are consolidated as a single chunk. Firstly, we performed an unsuccessful replication of a study reporting that the AB disappears when participants are instructed to remember the targets as a syllable. However, a subsequent experiment using easily combinable stimuli supported the model's prediction and showed a strongly reduced AB in a large group of participants. This result suggests that it is possible to avoid the AB with the right consolidation strategy. The skill-based approach allowed relating this finding to a general cognitive process, thereby demonstrating that incorporating this approach can be very helpful to generalize the findings of cognitive models, which otherwise tends to be rather difficult.

## Introduction

People rarely encounter a task that shares no similarities with tasks that have been done before. A smart strategy, therefore, would be to analyze which components of a task have been done before and which components are new, in order to be able to focus on the novel and more challenging aspects of a task. Such an approach could explain why people can learn certain simple new tasks with impressive speed. It can also explain why performance on some tasks is suboptimal: not because they are incapable of optimal performance, but because combining elements from a prior learning history leads to a suboptimal strategy. Furthermore, it may be a

**Data Availability Statement:** All relevant data are freely accessible using the following link: https://doi.org/10.5281/zenodo.5060680.

**Funding:** The authors received no specific funding for this work.

**Competing interests:** The authors have declared that no competing interests exist

crucial aspect of human cognition and be the reason why human behavior is so flexible and reliable.

The attentional blink (AB) paradigm is a good example of a simple but novel task. The objective for the participants in an attentional blink experiment is to identify and remember two targets amongst a stream of (to-be ignored) distractors. Although the task is fairly challenging given that items are presented at a rate of about 10 per second, participants only need to distinguish the targets from the distractors, remember the targets and report them at the end of the stream. Therefore, participants need to be provided with very limited instructions and require little time to execute this task properly. This impressive speed suggests that people can use skills they learned outside of the current context and apply them to the new context [1, 2]. In the example of the AB, in order to be able to remember the target stimulus the participants do not need to figure out from scratch how to remember a stimulus, they can simply use some sort of 'remembering' skill they already possess and use it to accomplish the AB task demands. This supports the notion that learning most new tasks does not require any new knowledge, it simply requires combining existing knowledge in a novel way.

In a previous paper [3] we created a cognitive model of the attentional blink (AB) based on this idea. This model was able to capture the most important aspects of the data reported in this paradigm and achieved this result using a novel and more human like modeling approach. This approach could provide certain improvements to areas in which models are currently lacking, which we will describe in the following section. The main objective in the current paper is to collect empirical evidence for the predictions made by this AB model on variations of this task. However, for reasons of clarity and completeness, we will first restate the reasoning behind the approach used to create the model and its contributions as well as the theoretical foundation of the AB model.

## The skill-based approach

At its core, the skill-based approach is a theory that explains the fast learning people are capable of when presented with a simple new task. This finding is hard to explain with cognitive models because they either require a long training session or a large amount of task-specific procedural knowledge specified by the modeler. The skill-based approach offers an explanation that does not require either. It does this by assuming that people can apply previously learned procedural knowledge, represented as skills, when these skills are useful in the context of the new task. Skill is a common word in the literature and has been used to refer to many different concepts. In this paper, skill refers to a collection of procedural knowledge that accomplish a certain general processing step and that can be used in multiple tasks. In our models, skills are represented by a set of operators (i.e., production rules) that can be used in different contexts by instantiating the variables depending on the context. Interestingly, when this theory is computationally implemented in this way, additional benefits to modeling in general become apparent because the strategy people employ to facilitate this type of fast learning (reusing of skills) seems to be a crucial element of human cognition underlying more characteristics of human behavior that have previously been difficult to capture by cognitive models.

The most striking example of such a characteristic is the impressive behavioral flexibility people possess. People are capable of performing a wide range of tasks while cognitive models developed to mirror this behavior are generally only capable of performing the specific task they are modeling. Presumably, this disparity is caused by the fact that people use general and reusable skills while cognitive models usually do not. In addition to underlying flexible and robust behavior in humans, the possibility of skill reuse also strongly limits the burden on

procedural memory. Instead of a different skill for every different context (e.g., a different skill for remembering names and remembering capitals), only a limited set of reusable skills needs to be stored (e.g., the same 'remembering' skill is used for names and for capitals). In summary, reusing skills allows people to behave in an efficient, flexible, and reliable way [2] and translating this strategy to cognitive modeling would also allow cognitive models to behave in a more efficient, flexible, and reliable way.

Additionally, the skill-based approach is an important addition to the value of cognitive architectures. Cognitive architectures are general modeling frameworks in which a large variety of tasks can be modeled (see e.g., [4] for an overview of cognitive architectures). Using a cognitive architecture to create a model offers two important advantages: (1) models will have a high level of cognitively plausibility since they are created within an empirically supported architecture and (2) models will be highly generalizable since they all operate within the same basic architecture. We believe that these two core aspects of cognitive architectures can be strongly improved by considering skill reuse when creating a new model (e.g., by applying some principles of the skill-based approach). Models that are created with such an approach will be more plausible since the pieces of procedural knowledge (i.e., the skills) are validated by using them in other (similar) tasks [3] and they cannot be (implausibly) (task-) specific since they need to be general enough to be reused. This will also increase the generalizability of a model since not only will the basic structure of human cognitive system be considered (the "architecture") but also how this system is used (the procedural knowledge).

Besides being important for the design of cognitive architectures, the skill-based approach may also improve the results of modeling efforts that do not make use of a cognitive architecture. Cognitive modeling is a tool used in many different fields answering widely varying questions. This presents a challenge with integrating this multitude of models into a single theory of cognition [5]. Possibly, the skill-based approach could aid integration of the many models because it allows researchers to more easily relate the mechanism they are modeling to the general field by explicitly defining the modeled mechanism as part of a more general cognitive process. Additionally, the skill-based approach could support the creation of a collection of skills from which modelers can draw from when building new cognitive models. This would be a huge step towards increasing the consistency between models of similar tasks based on the notion that models that perform the same processing steps should accomplish these processing steps in the same fashion (i.e., with the same skill).

We brought the skill-based approach into practice by creating a model of the attentional blink (AB) [3]. This model was created using the cognitive architecture PRIMs [6, 7]. PRIMs (which stands for primitive information processing elements) is based on ACT-R and has many of the same basic characteristics [5, 8]. Cognitive processing in both PRIMs and ACT-R revolves around information exchange within the central workspace by several modules capable of performing specific cognitive functions (e.g., the visual module is capable of visual processing). This way, a cognitive system gets built up from the modules capable of performing specific actions that can share the results of their actions with each other through the central workspace. The modules communicate in such a way that the result of the cognitive actions performed by one module can serve as input for the other modules. This allows for models to be created that are capable of performing a task from start to finish in many different fields [9–11]. The communication between the modules is controlled in PRIMs and ACT-R in largely the same way. In ACT-R this is done by productions and in PRIMs this is accomplished by operators, but they have generally the same functionality. A crucial advantage of using PRIMs over ACT-R is that PRIMs allows for operators (i.e., productions in ACT-R) to be organized into skills. A skill is a collection of operators that, combined, are capable of achieving a certain well-defined cognitive processing step within one model while still being general enough to be

reused in other models. The generalizability of skills allows for the same skills to be used in multiple models independent of which exact task is modeled. Additionally, the PRIMs architecture was developed with the intent of breaking up the relatively task-specific processing steps of ACT-R into more elementary and general steps. These elementary processing steps (the PRIMs) are central to the PRIMs architecture and facilitate creating reusable skills because the skills themselves are made up from general and elementary processing steps. This is the main reason why the PRIMs architecture is well suited for creating models based on skill-reuse.

## Skill selection in the attentional blink

Reusing skills is a vital part of human cognition and has many advantages. However, in some cases, this reuse of skills might have unintended negative consequences. Skills that work perfectly in some tasks might lead to sub-optimal performance in other (but highly similar) tasks even though the cognitive system is, in principle, capable of perfect performance [12]. The Stroop effect [13] might be the most famous instance of such sub-optimal skill selection. People are so used to reading, that the 'reading' skill is automatically triggered even when the task is to identify the color of a word (e.g. "red") rather than naming the word (e.g. the word "blue"), resulting in prolonged RTs in case of mismatch.

The attentional blink (AB) could be the inadvertent result of a comparable situation. The AB is an intensively studied paradigm in cognitive psychology [14, 15]. It refers to the finding that the second of two targets (referred to as T2) is often missed when it is presented in an interval of 200–500 milliseconds after the first (referred to as T1). However, when T2 is presented directly after T1, performance is not impaired and participants are able to identify T2 correctly most of the time. This Lag-1 sparing shows that people are, in principle, capable of remembering both targets, but that sub-optimal skill selection might lead to the performance impairment in identifying the second target for somewhat longer lags (e.g., at lags 2 or 3).

The crucial component of the sub-optimal performance may lie in the selection of the particular skill that accomplishes the consolidation of the targets in memory. Although there is no consensus on the exact mechanism behind the AB, memory consolidation has frequently been implicated to play a major causal role in this process [16] and many theories hold memory consolidation as the main factor underlying the AB [12, 17–20]. These theories differ in the fine details, however they all assume that the AB is the result of a similar two-stage process. This includes a first stage in which stimuli can be processed in parallel followed by a second stage in which only one stimulus can be consolidated into memory at the same time. According to these theories, this serial consolidation process forms the bottleneck responsible for the AB when T1 and T2 are presented in close temporal proximity but not immediately following each other. In these cases, T2 has to wait for T1 to be consolidated and, therefore, runs the risk of being overwritten in short-term visual memory by the following masking distractor, preventing T2 from being consolidated [21, 22].

Although the attentional blink is often conceived as a fundamental limit to human processing, several studies have reported various categories of manipulations that have led to substantial reductions and sometimes even complete eliminations of the AB, suggesting that the AB does not reflect a structural bottleneck. Some of these studies have manipulated the stimuli directly, e.g., the AB completely disappears when then the T2 is the participant's own name [23], however significant AB reductions have also been reported without manipulating the stimuli. One line of studies manipulating participants' attentional engagement in the task have reported a counter intuitive improvement to AB performance when participants were focused less on the primary AB task. These manipulations include playing music to the participants

and encouraging them to be distracted while performing an AB task [24], having the participants perform a concurrent secondary task [12], and distracting the participants with task irrelevant motion and flickering [25]. Additionally, a specific type of training has been shown to be beneficial for AB performance [26] and, finally, the existence of non-blinkers (individuals who do not display an AB) further questions the fundamental nature of the AB [14, 27, 28].

These manipulations reducing the AB without changing the stimuli strongly imply that strategy plays a crucial role in generating (and eliminating) the AB. We operationalize strategy as sub-optimal skill selection in this paper. The manipulations may have led to a reduced AB by successfully changing which strategy (and therefore which skill) participants were using to consolidate the targets into memory. It is our hypothesis that, instead of consolidating both targets separately, participants have been cued to consolidate both targets as a single chunk following these manipulations. This is similar to an earlier account, which states that the less engaged participants are unable to exert sufficient cognitive control to start consolidating immediately after encountering the first target [12]. The training may nudge participants towards using the chunked consolidation strategy and the non-blinkers might instinctively (or accidentally) employ this strategy. The crucial consequence of using the strategy of consolidating both targets as one chunk is that the bottleneck of stage-two processing as described earlier would not occur. Instead of consolidating T1 into memory as soon as it is identified (and therefore preventing T2 from being consolidated), T1 consolidation is postponed until T2 has been identified as well and both targets are consolidated together in one chunk.

Concrete evidence for the effect of strategy on AB performance was provided by an experiment performed by Ferlazzo and colleagues [29]. This paper reported the results of an experiment in which participants were either instructed to report the presented targets (which were always a vowel and a consonant) as two separate letters (the standard AB instructions) or to report them as a syllable. Interestingly, participants did not show an AB in the latter syllable condition. A possible explanation could be that participants in the syllable condition adopted a chunking consolidation strategy and thereby avoided the AB bottleneck, whereas the participants in the separate condition adopted the standard separate consolidation strategy and thus fell into the AB trap.

## Modeling the AB using the skill-based approach

In the paper mentioned at the start of the introduction [3] we investigated this effect of strategy on the AB by creating two versions of a cognitive model of the AB that only differed in their consolidation skill. The "consolidate-separate" version of the model consolidated the two targets as separate chunks into memory and the "consolidate-chunked" version of the model consolidated the two targets as a single chunk into memory. The model was created using the skill-based approach, which meant that instead of creating a model specifically for the AB, we composed the model from skills taken from other models. The skills used in both versions of the model were mostly identical, the only difference was that the consolidation skill used by the "consolidate-separate" version was taken from a model of a complex working memory task (in which participants consolidate every target separately) while the consolidation skill used by the "consolidate-chunked" version was taken from a model of a simple working memory task (in which participants consolidate multiple targets as a single chunk).

Both versions of the AB model were capable of capturing the data reported in the literature (see Fig 1). The "consolidate-separate" version of the model successfully showed the most important aspects of data reported in the AB literature: Lag-1 sparing, the AB itself and the steady performance increase on the later lags (Fig 1A). The "consolidate-chunked" version of the AB model, crucially, does not produce an AB. This version of the model avoided the AB

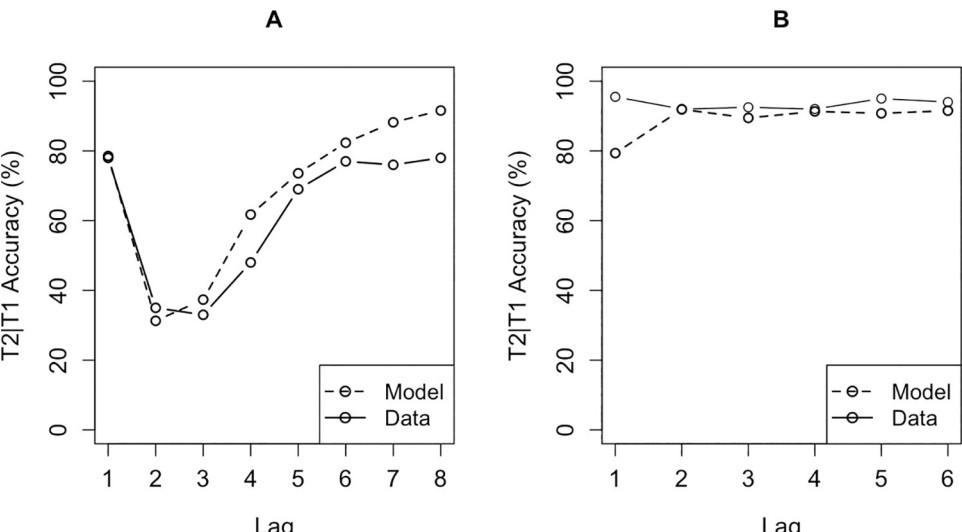

**Fig 1. Model fits for both versions of the AB model.** (a) the fit of the "consolidate-separate" model with data reported in a classic AB study [30]. (b) the fit of the "consolidate-chunked" version of the AB model with the data reported in the study by Ferlazzo and colleagues [29].

because memory consolidation is accomplished by a different consolidation skill. Instead of consolidating a single target into memory as soon as it is encountered, this consolidation-skill postpones consolidation until both targets have been detected and consolidates both targets as a single chunk into memory. We hypothesized that the syllable instruction condition of the study by Ferlazzo and colleagues prompted this consolidation strategy and, therefore, allowed the participants to bypass the AB. The "consolidate-chunked" version of the model indeed showed a good fit with the data reported in the syllable condition of the study by Ferlazzo and colleagues [29] (Fig 1B). Crucially, the only difference between both versions is the consolidation-skill, all other aspects of the model (e.g., model parameters) were held equal.

## Current study

The skill-based approach to model the AB did not only result in a model capable of capturing important aspects of the data reported in the AB literature but also constitutes a more naturalistic and human like modelling approach. This promising first step provided basic evidence for the potential of the skill-based approach. In the current study, we will attempt to collect additional empirical evidence for the central prediction of the AB model that the employed consolidation strategy greatly impacts AB performance.

We will do this by first attempting to replicate the previously mentioned study conducted by Ferlazzo and colleagues [29]. Secondly, we will perform an experiment centered around the same prediction using different targets than commonly used in AB studies. These two experiments allow us to test the effect of consolidation strategy on AB performance. Additionally, it provides an opportunity to test the flexibility of the models created by the skill-based approach, because the task demands of the second experiment are slightly different compared to the original AB task, requiring small adjustments to the model.

## Experiment 1

Experiment 1 was a replication attempt of the experiment in the study conducted by Ferlazzo and colleagues [29] in which the targets (a vowel and a consonant) had to be reported as a

single syllable. The goal was to verify the original findings and to create a complete data set which would allow for a more detailed model fit. Although we did not carry out an exact replication of the experiment reported in the original paper, the results should be comparable since the crucial manipulation was identical in both studies.

## Method

**Experimental setting.** The research took place in the lab of the Artificial Intelligence department of the Bernoulli Institute at the University of Groningen. The experiment was run and programmed with OpenSesame [31] using the backend PsychoPy [32] on a MacOS computer. The participants were seated approximately 0.5 meter away from the computer screen, which was a 24 inch LCD Benq XL2420-B with a refresh rate of 60 Hz.

**Participants.** All 18 participants (10 female, aged 18 to 25, mean = 21.2 years) who took part in the experiment were students of the University of Groningen and received a financial compensation of 8 euros for their participation. The sample size was based on the large effect size reported by Ferlazzo and colleagues [29]. Additionally, a power analysis was performed utilizing the method provided by [33]. This analysis was based on the effect size and standard deviation reported by the original study and indicated that 14 was the recommended sample size for those values. Finally, prior to the experiment, participants signed an informed consent form and ethical approval was acquired from the Research Ethical Review Committee of the University of Groningen.

**Stimuli.** On every trial a sequence of twenty stimuli was presented consisting of 18 distractor stimuli (digits) and 2 target stimuli (letters). The distractors could be any digit with the exception of 1, 5, and 9. They were randomly drawn with the single rule that two subsequent distractors could not be identical. The targets on every trial consisted of a vowel-consonant pair (creating a syllable). The order of presentation was random but the frequency of appearance was balanced; on half of the trials the vowel was presented first (i.e., as T1), on the other half of the trials the consonant was presented first. The vowel on every trial was randomly drawn from a collection of four vowels: 'A', 'E', 'I', and 'U'. The consonant on every trial could be any consonant (except for 'S', 'Q' or the semi-vowel 'Y'). The stimuli were presented in white at the center of the screen on a black background. The font used for both the targets and distractors was 'droid sans mono' which is the default font used by OpenSesame. The size of both the distractors and the targets was about 1˚ of visual angle.

**Procedure.** The experiment was set up with a between-subjects manipulation of instruction. The participants were instructed to either remember the two target letters as two separate letters (the 'separate' condition) or as a single syllable (the 'syllable' condition). Participants in either condition were not aware of the other condition. Participants were randomly assigned to a condition, which due to a technical oversight led to a small imbalance between the conditions. At the end of data collection, 10 participants had been assigned to the 'syllable' condition and 8 participants had been assigned to the 'separate' condition. Additionally, the serial position of the second target (T2) relative to the first target (T1) was varied (referred to as lag). The lags included in this study were lag 1, 2, 3, 4, 5, and 6. All 6 lags were presented 70 times (420 experimental trials in total).

Before the experiment started, participants received a short verbal instruction from the experimenter. This verbal instruction was given in addition to further written instructions on the computer screen which participants could read at their own pace. The verbal instructions were given because participants' understanding of the instructions was a crucial part of the experimental manipulation. Finally, participants performed 18 practice trials, 3 trials per lag.

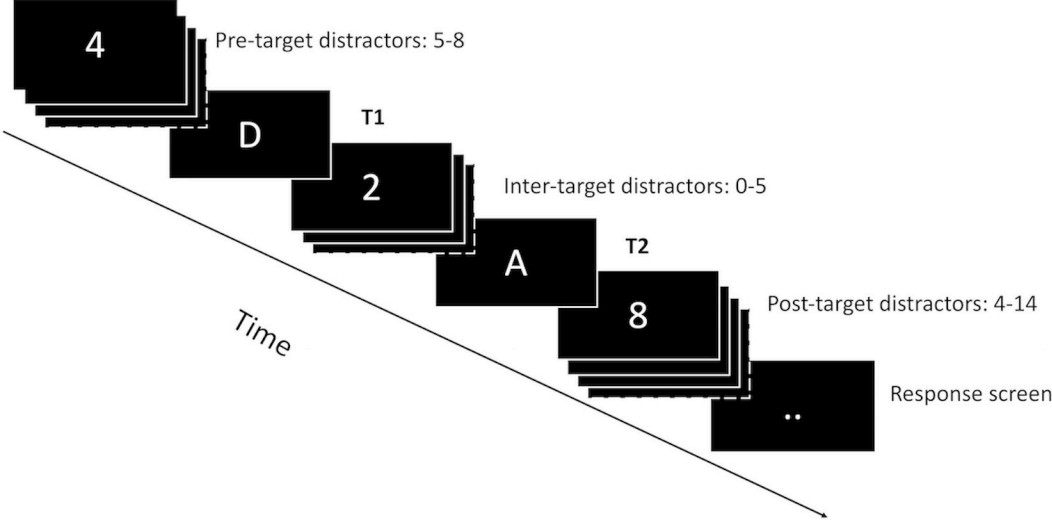

**Fig 2. Schematic representation of a trial in Experiment 1.**

The participants received feedback about their performance during the practice trials, but they did not receive any feedback during the experimental trials.

All trials in the experiment consisted of a rapid serial visual presentation (RSVP) stream containing 20 items presented at a rate of 10 Hz (see Fig 2). The RSVP stream was always preceded by a fixation cross presented for one second in the center of the screen. The stimulus onset asynchrony (SOA) in this study (100 ms) was longer than was reported in the original study (80 ms). This was decided after an unsuccessful pilot study with low accuracy indicating that an SOA of 80 ms was too fast in the current design. One reason for this discrepancy with the original study could be that the targets were more easily distinguishable from the distractors in the original study (e.g., because of the font), but this cannot be verified from the reported information. Finally, at the end of every trial, two white dots appeared in the center of the screen prompting the participants to type in the two letters they had seen during the RSVP stream. This response screen was identical in both conditions. The participants provided their answers with the letter keys on the keyboard without time pressure. Participants required approximately 45 minutes to complete the experiment.

## Results

The main goal of Experiment 1 was to replicate the original findings as reported by Ferlazzo and colleagues [29] regarding the effect of instruction on AB performance. They reported a strongly reduced AB in the 'syllable' condition. We conducted a highly comparable experiment and thus expected that the participants in the 'syllable' group would show a strongly reduced AB while the participants in the 'separate' group would show a standard AB.

However, the data do not support this hypothesis. As can be seen in Fig 3A, there were no large differences in T2|T1 accuracy between the two conditions at any lag. T2|T1 accuracy refers to the T2 accuracy on trials in which T1 was correctly identified. The largest difference exists at Lag 2 where average performance in the 'syllable' condition was slightly higher than in the 'separate' condition. However, this difference was not statistically significant as tested with a logistic linear mixed effects regression model ($\beta = 0.22$, $SE = 0.42$, $z = 0.5$, $p = .59$). Furthermore, the data from our 'syllable' condition differs strongly from the original data and does not show the strong AB reduction reported in the original study. In short, the data does not

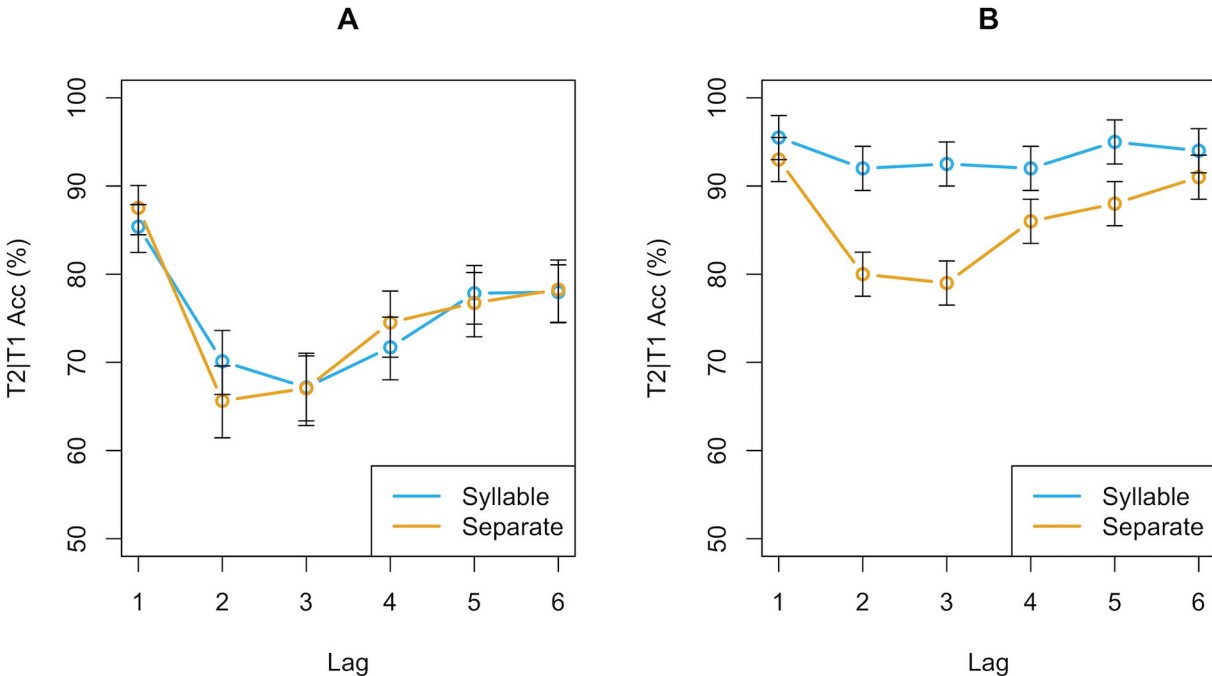

**Fig 3. Comparison of replication and original study.** (a) the mean T2|T1 accuracy per instruction condition over the six lags in our study. (b) the mean T2|T1 accuracy of the original study by Ferlazzo and colleagues [29]. In contrast to the original, performance in the 'syllable' condition (blue line in 3a) did not significantly differ from the AB in the 'separate' condition (orange line in 3a) in our study. Additionally, the difference in general accuracy between our data and the 'separate' condition of the original study (orange line in 3b) implies a difference in difficulty level of the two studies.

support the conclusion that the AB magnitude differs significantly between the two instruction conditions and thus failed to replicate the findings of the original study.

## Discussion

Ferlazzo and colleagues [29] reported the results of an AB experiment in which participants showed a strongly reduced AB when instructed to report the two targets as a syllable. We explained these results by assuming that these participants employed a different consolidation strategy than commonly employed in the standard AB task. Instead of consolidating both targets separately into memory, they may have consolidated both targets as one chunk and therefore managed to avoid the bottleneck of memory consolidation thought to underlie the AB.

In Experiment 1 we attempted to replicate these results. The design of Experiment 1 was highly similar to the design of the original study involving the same manipulation of instructing the participants to report the targets either as a syllable or as separate letters and used similar stimuli. Nevertheless, this experiment failed to replicate the results of the original study: there was no significant difference in AB magnitude between the two instruction conditions.

The main reason for the failed replication may be that we were unable to effectively manipulate the participants' consolidation strategy. This could perhaps have been caused by subtle differences in how the instructions were phrased. Additionally, the participants in our sample might have been less sensitive to the syllable instructions. Possibly, due to differences in linguistic and cultural background, the concept of a syllable may not have been as clear to our Dutch participants compared to the original Italian participants, and, therefore, may have been ineffective in spurring the chunked consolidation strategy (Ferlazzo, personal communication, June 26, 2019). Note also that the SOA was 20 ms longer than in the original study,

which may have played a role in the effectiveness of inducing the chunked-consolidation strategy. In addition, the targets in the original study may have been substantially easier to distinguish from the distractors, which could have improved performance and facilitated in the chunking of the targets. This possibility is supported by the fact that the participants in the 'separate' condition of the original study outperformed the participants in both of our conditions. Note, however, that the reduced AB in the original study could not simply have been an artifact of the stimuli used in the original study. The original study also included a 'separate' condition which showed a mostly standard AB (as can be seen in Fig 3B).

Although the original results were not replicated, it seems unlikely that those results were merely a chance finding. There were small differences between the original study and the replication which might have prevented us from successfully manipulating the participants' consolidation strategy. Furthermore, the original paper reports two additional experiments that both supported the results of the experiment we attempted to replicate. To conclude, although the failed replication suggests that the manipulation of the original study does not reliably result in the same effect, one may also argue that it does not categorically reject the original results but rather shows how difficult it is to manipulate participant strategy. Therefore, we conducted Experiment 2 involving a stronger manipulation of participant strategy in order to accurately test the predictions of our AB model.

## Experiment 2

The manipulation we included in the design of Experiment 1 seemed to be insufficient to cue participants to use the chunked consolidation strategy. An additional manipulation enforcing this strategy is thus necessary to create a suitable data set to test the predictions of our AB model. Therefore, Experiment 2 was set up, containing targets that were expected to promote chunking in addition to the instruction manipulation.

### Method

**Participants.** In total, 82 participants (45 female, average age: 20.9) took part in the experiment. All participants were students of the University of Groningen who received a financial compensation of 8 euros for their participation. Two participants were removed from the final data analysis because too many trials had to be excluded (see 'data preparation' below for more details). The sample size was increased in response to the small effect of the instruction manipulation found in Experiment 1. Ethical approval was acquired from the Research Ethical Review Committee of the University of Groningen and participants signed an informed consent form before taking part in the study.

**Stimuli.** The same experimental setting and apparatus was used as in Experiment 1. On each trial a sequential stream of twenty stimuli was presented, including multiple distractor non-targets and one or two targets. The distractors were letters, randomly drawn with replacement, with the additional constraint that two sequential stimuli were never identical. The targets in this study were chosen in order to promote chunking. They consisted of corners of a square, as used in a study by Akyürek and colleagues [34], shown in Fig 4A. Every corner of the full square was 7 pixels wide and both the horizontal and the vertical side were 23 pixels long with 8 pixels of white space in between two neighboring corners. The white space in between the corners gave the impression that the corners were distinct but, because of the total configuration, still part of a bigger figure (e.g., forming a square). On most trials, two targets were presented consisting of one or two corners. The exact make up of these two targets was determined randomly with the single rule that there could be no overlapping corners (e.g., if T1 consisted of the bottom left and the bottom right corner, then T2 could not contain any of

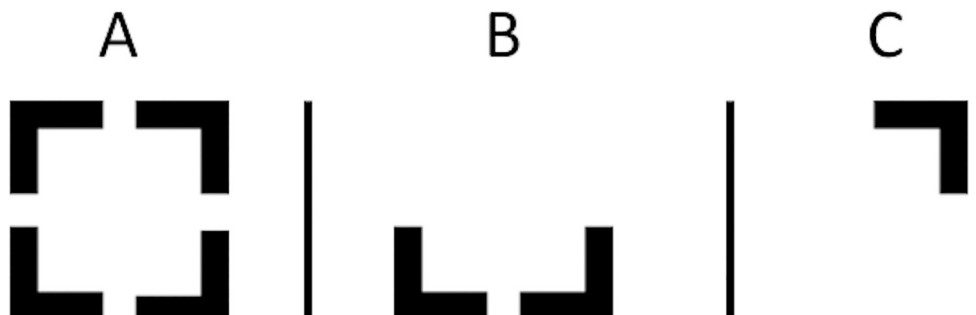

**Fig 4. Examples of the stimuli used in the study.** (a) the full square that was used as the base for the creation of the targets. (b) an example target of the maximum size (i.e., 2 corners). (c) an example target with only one corner. The two targets in one stream could consist of any combination of these targets as long as there were no overlapping corners.

these corners), see Fig 4B and 4C for examples of targets. Both the targets and the distractors appeared in a similar size of around 2 degrees of visual angle on the screen. The font used for the distractors was 'droid sans mono'.

**Experimental design and procedure.**   The experiment was set up with a between-subjects manipulation of instructions. Participants in the 'separate' group were instructed to report both targets separately whereas participants in the 'combined' group were instructed to report the two targets as a single unit. Participants were not aware of the existence of the other condition. Additionally, the lag of T2 was varied. In this study lags 1, 2, 3, 4, 6, and 8 were included in the design. All 6 lags were presented equally often (50 trials per lag) and the order was determined randomly (without replacement).

Before the experiment started, verbal instructions were provided in addition to written instructions on the screen in a similar manner as in Experiment 1 to ensure adequate understanding of the task. Additionally, participants performed 24 practice trials with two targets (4 for each lag) and 4 trials in which only one target was presented to further ensure correct understanding of the task (28 in total).

The experiment consisted of 378 trials in total (including practice) and took around 45 minutes to complete. Every trial in the experiment consisted of an RSVP stream of 20 items with a presentation rate of 12 Hz (i.e., every item was on screen for 83.3 ms) preceded by a fixation cross which was on screen for 1 second. The presentation rate was slightly faster than in Experiment 1 and most AB tasks (frequently presented at 10 Hz) to make the task sufficiently challenging for the participants. On most of the experimental trials (300 out of the 350 experimental trials) two targets were presented. On the remaining 50 trials only one target was presented and the place of the T2 was taken up by an additional distractor.

Finally, at the end of every trial one or two response screens appeared (depending on the instruction condition). The response screen displayed a 4 by 4 grid with all response options (16 options in total). These response options consisted of all 15 possible targets and an option to indicate that there was no second target present or that the target in question was missed. The response option screen was identical in both conditions. Each response was given by pressing the appropriate key on the keyboard that was associated to it (a key was displayed underneath every response option). There was no spatial regularity between the location of the response options on the screen and the location of the keys on the keyboard. The keys with which the participants responded were the numerical keys at the top of the keyboard and the letters 'Q' up to 'T'. The 'Enter' key was used to indicate that the participants had not seen the target. There was no time constraint on the responses.

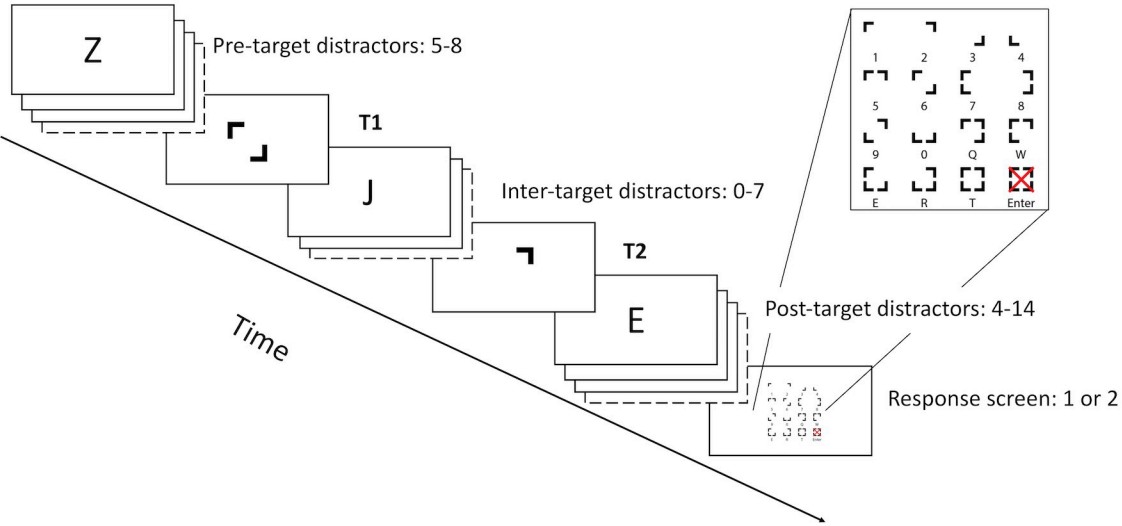

**Fig 5. Schematic overview of a single trial with two targets.** Every trial started with the presentation of a fixation cross for one second (not shown). The presentation rate of the stimuli was 12 Hz with no inter stimulus interval.

Note that participants in the 'separate' group gave two responses while participants in the 'combined' group were only required to give a single response (Fig 5). This difference may have resulted in some of the participants in the 'separate' condition to also attempt to remember the order of the two targets. They were not instructed to report the two targets in the order in which they appeared and the feedback they received during the practice trials also did not depend on this order. However, the participants were also not explicitly instructed to report the targets in any order, so it is possible that some participants did attempt to remember the order of the targets. This might have made the task slightly more difficult for some of the participants in the 'separate' condition. However, we expected that this difference would only influence general accuracy and not AB magnitude. Furthermore, we took measures to limit the influence of this difference which are described in the next section. The crucial aspect of these measures is that the order was not relevant in either condition during data analysis.

**Data preparation.** The stimuli used in the study and our hypothesis required a somewhat different way to collect the participants' responses than is common in AB tasks. The main difference between our study and most AB tasks is that the participants in the 'combined' instruction group only provided a single response. Because of this, we needed a method to be able to distinguish between T1 and T2 accuracy based on this single response. Additionally, in order to match the difficulty of responding correctly in both conditions of our study, the two responses given in the 'separate' condition had to be treated as a single response, ignoring the order of the responses, similarly to the 'combined' condition. To resolve these two issues, we transformed our data in the following way. Firstly, we combined the two responses given on every trial by the participants in the 'separate' condition into a single response. After this step, we treated the data from both conditions in the same way. Extracting the T1 and T2 accuracy from these single responses was subsequently done by comparing the presented T1 and T2 on a trial to the response given on that trial. If the response included all the corners belonging to one of the targets, this target was counted as correct. On the other hand, if the response did not include all the corners presented as one of the targets, this target was considered to be incorrect. Trials in which the participants reported more corners than were presented were excluded from the data analysis (7% of trials). Exclusion was chosen over simply judging the

response incorrect because it cannot be retroactively determined whether the error should be attributed to T1 or to T2. Finally, as mentioned before two participants were completely excluded from the analysis because more than half of their trials were excluded after this step, leaving a total of 80 participants in the analysis.

## Cognitive model

**Original cognitive model.** A large portion of the model has already been created as part of a previous effort [3]. For clarity, we will first repeat and summarize the methodology of that work before describing the adjustments we made in order to adapt the model to the current task demands.

The initial cognitive model was constructed using the skill-based approach as outlined in the introduction. This meant that instead of creating a model specifically for the attentional blink it was assembled from general skills, mirroring how participants would approach a new task. Based on previous research and other models of the AB, we identified four basic skills required to successfully perform an AB task. These four basic skills were created by developing three models of other tasks that also made use of one or more of these skills. These three models were: (1) a visual search model, (2) a model of a complex working memory (CWM) task, and (3) a model of a simple working memory (SWM) task. Complete descriptions of the three basic models and more details on the modeling methodology can be found in the previously mentioned paper, where we also show that the models fit the appropriate experimental data for the memory tasks [3].

After creating the three basic models, all the building blocks needed to create the final AB model were present and the AB model was assembled in the following way. The visual search model provided the first skill for the AB model: the "search" skill, used to distinguish targets from distractors in the AB stream. The simple working memory model and the complex working memory model provided the other skills. Firstly, the CWM model supplied the "consolidate-separate" skill, consolidating one individual item into memory. The reason for using this strategy is that in CWM experiments, the items that have to be memorized are interleaved with another task. This consolidation skill was used for the version of the AB model that modeled the standard consolidation strategy during AB tasks. The model of the SWM task provided the alternative consolidation strategy. This consolidation skill is capable of consolidating multiple items as a single chunk into memory, based on evidence in the literature that people do indeed use chunking strategies in such experiments [35]. This skill was used for the alternative AB model. The CWM and SWM models provided two additional skills, one responsible for retrieving the consolidated items from memory (the "retrieve" skill) and the other responsible for responding with the retrieved items (the "respond" skill). Thus, both working memory task models and both versions of the AB models used the same "retrieve" and "respond" skill.

**Modified cognitive model.** A modification to one of the skills described above had to be made in order to align the model with the demands of the task used for Experiment 2. This new task is largely the same as the previous task with the exception that the stimuli were different. Because of this, the visual search skill had to be instantiated differently.

Instantiation plays a crucial role in facilitating skill reuse because it allows for the task-specific information used by a skill to be adjusted without changing the procedural rules (i.e., the operators). In PRIMs models, skills often contain a certain number of variables which can be specified differently depending on the concrete task the model will be performing. For example, the same visual search skill is used for the model of Experiment 1 and Experiment 2, but the variable *"distractor-type"* (specifying the nature of the distractors) is instantiated differently

(among other variables). In the model of Experiment 1 this variable is instantiated as "*number*" and in the model of Experiment 2 this variable is instantiated as "*letter*". Because all task-specific symbolic knowledge is represented by such variables it becomes possible to reuse skills in different contexts without changing the skills themselves.

In addition, a variable can also be instantiated with another skill. The main advantage of instantiating variables with a skill is that for every skill it can be specified which skill to perform after it. This is crucial for skill reuse since it allows for the general skills to be carried out in any order (e.g., in some tasks it is required to immediately respond after retrieving the correct answer while in other tasks it might be required to perform some additional processing before responding, but this does not change the skill itself). We used this characteristic to adapt the four basic skills we created according to the demands of the AB task. For example, in the visual search skill a subskill determines how to get to the next stimulus. In a standard visual search task this is done by an eye movement to the next unattended stimulus, but in the AB task it is to wait for a new stimulus in the same location.

The changes to the visual search skill had one important consequence for the performance of the model. Instead of using a declarative memory retrieval as selection criterion to distinguish between letters (the targets) and numbers (the distractors), a perceptual judgement was used as criterion to distinguish between the corners of a square pictured in Fig 2 (the targets) and letters (the distractors). The main impact of this change is that the perceptual judgment is faster than a memory retrieval.

After modifying the visual search skill, all skills were put together in the AB model which carried out the task in the following way. If a letter (i.e., a distractor) is presented, the model ignores this stimulus and waits for the next. When a corner of a square (i.e., a target) is encountered, the model switches to one of the "consolidation" skills (depending on which version is run). If the "consolidate-separate" skill is performed, the model instantly consolidates the target into memory and no other operator can be executed for, on average, 200 milliseconds (the imaginal delay parameter in ACT-R and PRIMs), possibly leading to an attentional blink. The duration of this period varied between 50 and 350 ms, randomly determined using a uniform distribution. If the "consolidate-chunk" skill is performed, the model postpones consolidation of the target until the second target is encountered and keeps performing the task normally, such that no blink occurs in this version of the model. After all stimuli have been presented, the model uses the "retrieve" skill to retrieve the consolidated items from memory and, finally, uses the "respond" skill to respond with the retrieved items.

The underlying cause for the AB in our model is very similar to an AB model created in ACT-R [12] in the sense that the AB is caused by a wrong consolidation decision and not by fundamental information processing limits. The crucial aspect of the consolidation decision revolves around when the contents of the imaginal buffer (i.e., working memory) are committed to longer term storage. In our model this moment of encoding is directly linked to the consolidation skills. The 'consolidate-separate' skill encodes an item into longer term memory as soon as it encounters a target, while the 'consolidate-chunked' skill will only start consolidation when both targets have been detected. That the encoding moment can depend on the context is suggested by AB studies that use three sequential targets [36, 37]. In these studies, the third target (presented at the 'lag 2' position in normal AB studies) is reported just as frequently as the first target, completely eliminating the classic AB effect. These results support the idea that the moment of memory consolidation varies depending on context or perhaps strategy. In this conception, the AB is caused by the fact that the 'consolidate-separate' skill is the default consolidation skill for most participants in the AB context and that it can be avoided when participants are compelled to use the 'chunked-consolidation' skill.

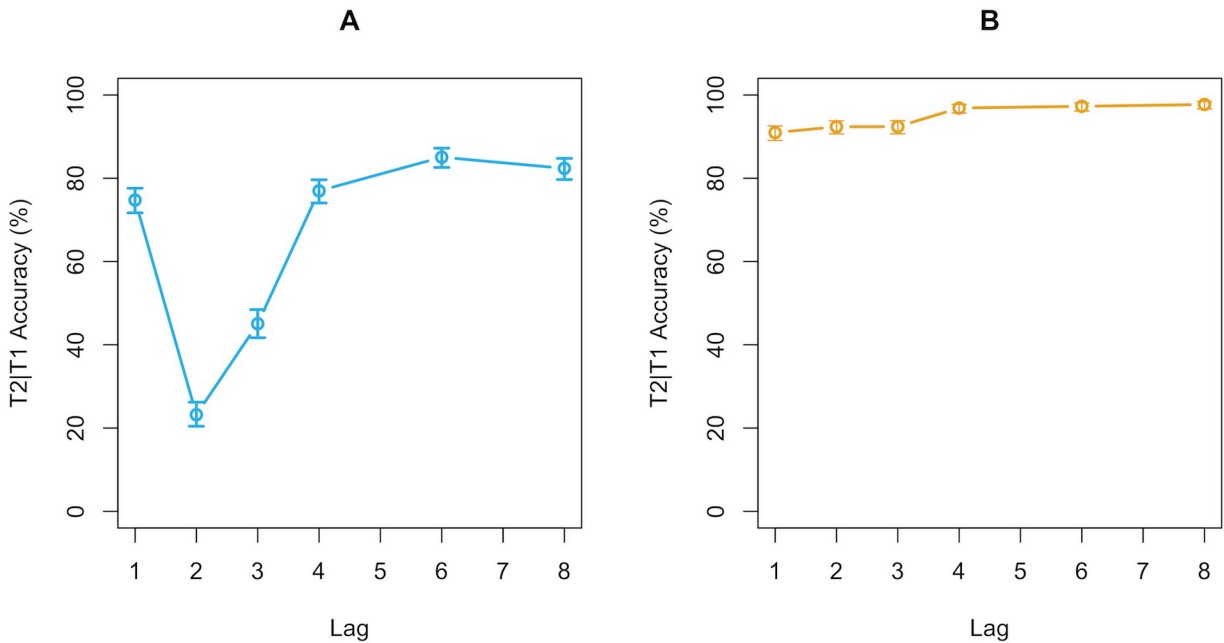

**Fig 6. Model predictions for both version of the modified AB model.** The predictions of the 'consolidate-separate' version of the model include a mostly standard AB with the deepest point at lag 2 (a). The predictions of the 'consolidate-chunked' version include no AB and a very high overall performance level (b). Confidence intervals were calculated using the Agresti-Coull method [38].

The predictions of the model (see Fig 6) are similar to the predictions by the original PRIMs model shown in Fig 1. These predictions were made by running both versions of the model 20 times with 500 trials per run for a total of 20000 trials (10000 per version).

## Results

**Effect of instruction.**   Our main hypothesis concerned the effect of instruction on performance in an AB task (Fig 7). We predicted that the participants in the 'combined' condition would show a smaller AB than the participants in the 'separate' condition. We chose to define AB magnitude by the slope with which T2|T1 accuracy improved over the lags. This method was recommended by MacLean and Arnell [39], especially for studies investigating modulations of the AB, and has been used before in a similar fashion (e.g., [40]). The main advantage of this method over taking difference scores (e.g., Lag 8 –Lag 2) is that it factors in all intermediate lags, not just the extreme lags. Furthermore, this method allows for quantification of the effect, since it not only provides information about whether the slopes are significantly different (i.e., the p-value), but also provides an indication of how different the slopes are (i.e., the effect size).

To test our hypothesis, we fitted a logistic generalized linear mixed effects model on the T2| T1 accuracy data. This was done with the statistical software 'R' [41, 42] using the package 'lme4' [43]. P-values were extracted with the package 'lmerTest' [44] using the Satterthwaite method [45, 46].

Our final model, testing the effect of instruction on T2|T1 accuracy, included two fixed effects (Instruction and Lag) and a random intercept for subject. Instruction was a categorical factor with 'separate' as the reference level. Lag was a numerical factor starting at Lag 2. Lag-1 trials were not included in this model because of our definition of AB magnitude (see above). The analysis revealed no significant main effect of Instruction ($\beta = 0.0.18$, SE = 0.35, z = 0.5,

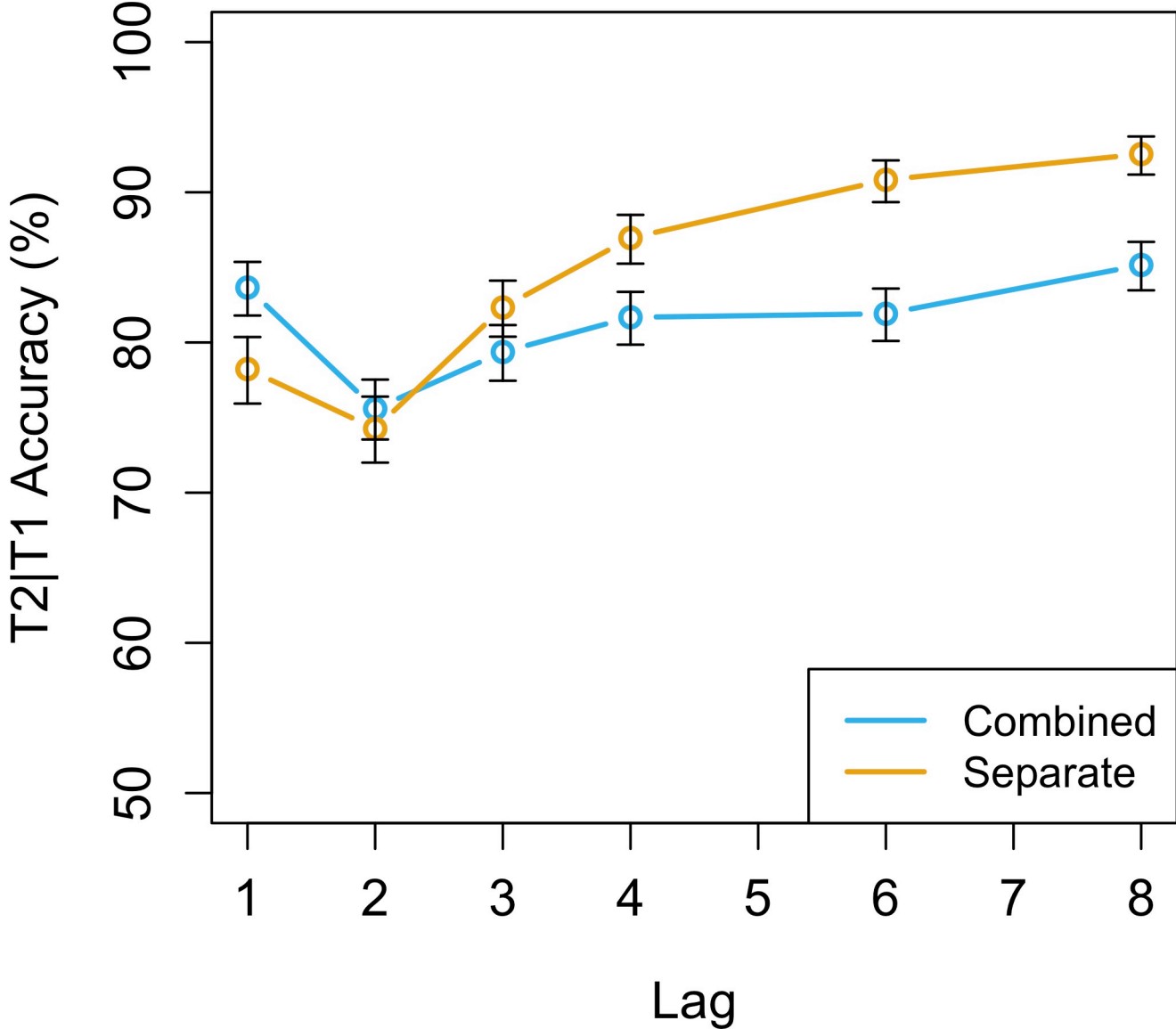

**Fig 7. T2 accuracy as a function of instruction and lag.** T2 accuracy in the 'combined' condition (blue line) has a significantly smaller slope over lag than T2 accuracy in the 'separate' condition (orange line). Confidence intervals were calculated using the Agresti-Coull method [38].

p = .6) indicating that there was no difference in T2 performance between the two instruction conditions at the first level of Lag (i.e., Lag 2). The analysis did reveal a significant main effect of Lag ($\beta$ = 0.29, SE = 0.02, z = 15.9, p < .0001) with the positive beta coefficient of 0.29 indicating that T2 performance improved on later lags. Finally, the analysis revealed a significant interaction between Lag and Instruction ($\beta$ = -0.14, SE = 0.02, z = -5.6, p < .0001) implying that the slope of Lag is less steep in the 'combined' condition which indicates that the participants in the 'combined' condition showed a relatively smaller AB.

We, additionally, performed a post-hoc analysis of the unexpected difference between the two conditions at lags 4–8. In order to perform this analysis, we ran the same model but with Lag as a categorical factor instead of as a continuous variable. Entering Lag as a categorical factor allowed us to test for an interaction between Lag and Instruction for every level of Lag

separately instead of testing for an interaction over all levels of Lag collectively. This analysis revealed a significant interaction between Instruction and Lag at lag 4 ($\beta$ = -0.34, SE = 0.14, z = -2.3, p = .02), lag 6 ($\beta$ = -0.74, SE = 0.15, z = -4.9, p < .0001), and lag 8 ($\beta$ = -0.67, SE = 0.16, z = -4.2, p < .0001) indicating that performance was significantly lower in the 'combined' condition at these lags.

In addition to analyzing the T2|T1 accuracy, we also conducted statistical tests on the effect of Instruction and Lag on T1 performance in a similar fashion. Our final model included Instruction and Lag both as categorical factors and a random intercept for subject. Average T1 performance differed slightly per condition: 94% in the 'separate' condition compared to 91% in the 'combined' condition. However, this difference did not reach significance at any of the lags (all ps > .05).

Finally, we analyzed the performance on the trials where only one target was presented. Participants performed very well on these trials, correctly reporting the 'T1' in 96% of the trials. Performance on these trials was almost identical across the conditions with 97% accuracy in the 'separate' condition and 96% accuracy in the 'combined' condition.

**Model predictions for the effect of instruction.** The model predicted different performance patterns for the participants in the 'combined' condition and in the 'separate' condition. In particular, it predicted that the participants in the 'separate' condition would show a strong blink characterized by a serious decline in performance at Lag 2 and a quick recovery at Lag 3. In contrast, the model predicted that the participants in the 'combined' condition would not show an AB, performing close to ceiling at all lags. Although the model predictions were in the correct direction (there was a relatively smaller blink in the 'combined' condition), the difference between conditions was much smaller than anticipated by the model. Therefore, the model predictions regarding the effect of instructions were not fully supported by the data.

The model also predicted that the targets that were used in Experiment 2 (see Fig 4) would have an impact on performance. Because it was relatively easy to distinguish the targets from distractors in comparison to the ones typically used in regular AB designs, the task in Experiment 2 did not require a memory retrieval. The first step done by the model (i.e., detecting T1 from the stream of distractors) could therefore be accomplished faster, which subsequently caused the model to be faster in completing consolidation of the targets. This meant that the entire AB occurred slightly earlier, resulting in the atypically quick recovery in model performance at lag 3. This prediction was supported by the participant data: the AB was deepest at lag 2 and recovered quickly at lag 3.

Although the significant difference in AB magnitude between the two groups shows an influence of the instruction manipulation, its impact was not as large as predicted by our model. This weaker than predicted effect of the experimental manipulation may have been caused by an only partial success of the strategy manipulation. The unpredictable nature of strategy manipulations, as suggested by the results of Experiment 1, implies that the instruction manipulation may have had a different effect on individual participants. Some participants in the 'combined' condition may have still used the more natural separate consolidation strategy, furthermore, some participants in the 'separate' condition may have been prompted by the nature of the targets to adopt the chunked consolidation strategy.

Additionally, the data analysis revealed an unexpected difference between the instruction conditions at the later lags (lags 4–8). Before data collection, we expected that the 'separate' condition may be slightly more difficult because participants might attempt to also remember the order of the targets. However, the data analysis showed that this relationship was the other way around: rather than better, performance was worse in the 'combined' condition. This finding is remarkable because the 'combined' condition should always be less difficult than the

'separate' condition. After all, if it is possible to report the two targets separately, it should also be possible to report them as one combined unit.

In order to better understand these two issues, a closer look at the individual performance patterns of the participants in our study is required. To accomplish this, we performed a cluster analysis. This method allowed us to extract the most commonly occurring performance patterns in the data and to group similarly performing participants together based on the observed patterns of behavior.

**Cluster analysis.** Cluster analysis is a collection of data driven statistical methods able to create groups of participants within a single data set. In our case, we wanted to group the participants in our experiment based on their T2|T1 accuracy for each lag. Many different clustering methods exist, differing in complexity and their applicability to certain data sets. We used the k-means clustering algorithm, which groups the data based on various sets of numbers (vectors) with which the squared distance from one of these sets is minimized for all sets that were observed (in our case, each observed set refers to a single participant). The cluster analysis was carried out with the R-package 'factoextra' [47].

Before a k-means cluster analysis can be performed, the $k$ (i.e., the number of clusters resulting from the analysis) has to be determined. We accomplished this using a common approach actualized in the R-package 'NbClust' [48]. This method calculates 30 different measures calculating the optimal number of clusters, with the most optimal number of clusters being the number that received the most support from all 30 measures. For our data, 12 of these 30 measures suggested that 3 was the optimal number of clusters. The next most supported number of clusters was 2, with 5 measures supporting it. Therefore, we performed a k-means cluster analysis on the T2|T1 performance data for all lags included in our study with $k = 3$. This means that our cluster analysis will result in three groups of participants with different performance patterns.

The cluster analysis resulted in three differently sized groups of participants who showed distinctly different T2|T1 accuracy patterns (see Fig 8). The three clusters together explained 76.8% of the total variance in the data. The first and largest cluster consisted of 52 participants of which 28 came from the 'combined' condition and 24 from the 'separate' condition. It contained the participants who performed remarkably well in our study, performing at approximately 90% accuracy and showing no or little AB. The second cluster contained 16

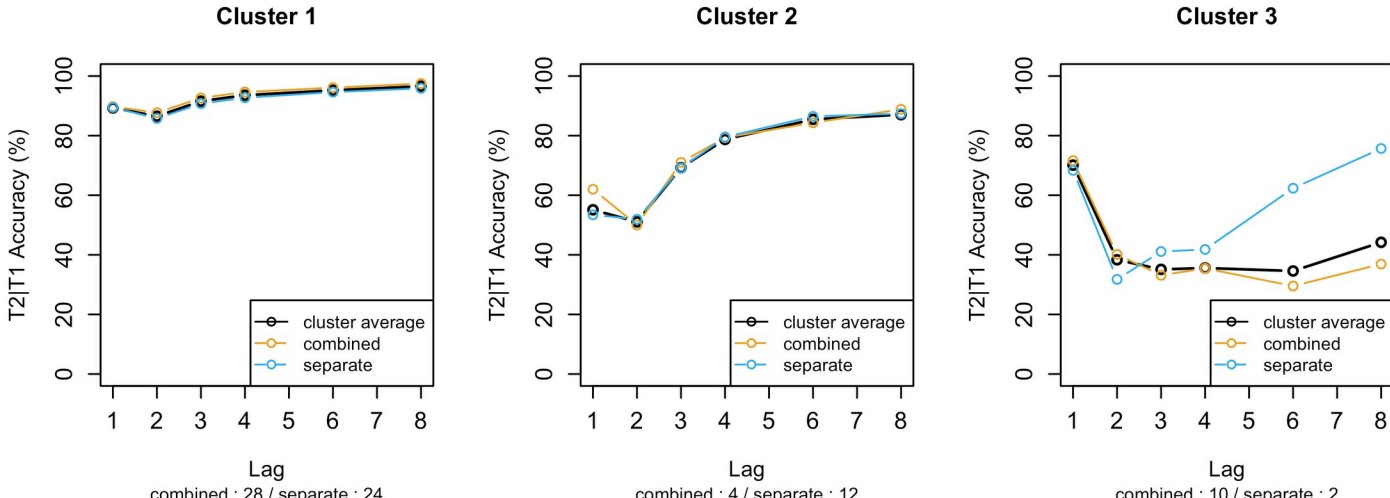

**Fig 8. The three clusters present in the T2|T1 accuracy data.** Indicated underneath the x-axis label are the number of participants included in this cluster per condition. Additionally, the dashed lines depict the cluster averages separately per instruction condition.

participants of which 12 were in the 'separate' condition, showing a mostly classic AB pattern, but demonstrating little lag-1 sparing. The final and smallest cluster contained 12 participants of which 10 had been in the 'combined' condition and showed a performance pattern that was quite different from the previous two. Whereas these participants performed relatively well at Lag 1 with 70% accuracy, they performed quite poorly at the subsequent lags, hardly reaching 40% accuracy.

An interesting aspect of the results of the cluster analysis was that participants in the 'separate' condition seemed more likely to be assigned to Cluster 2 while participants in the 'combined' condition were more likely to be assigned to Cluster 3. Using a Pearson's chi-squared test on the distribution of participants from the two conditions assigned to either Cluster 2 or 3, we found a significant difference ($\chi2$ (1) = 7.1; p = .008), confirming that condition was an important factor in determining which cluster a given participant was assigned to. A comparable result was achieved when the distribution of participants from both conditions over all three clusters was tested ($\chi2$ (2) = 9.5; p = .009).

In addition to examining the differences between the instruction conditions across clusters we can inspect the differences between the conditions within a cluster. Fig 8 additionally shows the average performance of participants assigned to each cluster separately for both conditions. It reveals that the participants in both conditions performed very similarly to the cluster average in Cluster 1 and Cluster 2, indicating that the cluster analysis was successful in grouping comparable participants in those clusters. However, the participants in Cluster 3 differ in one crucial aspect. The participants in the 'separate' condition show a performance pattern characterized by a deep and long AB including a return to normal performance levels at the later lags, while the participants in the 'combined' condition show consistently poor performance. This supports the idea that the instruction manipulation led participants to adopt different strategies. Specifically, it appears that the participants in the 'combined' condition rarely used the standard 'consolidate-separate' skill indicated by the very small number of participants showing a regular AB. However, it also suggests that quite some participants in the 'separate' condition used the 'consolidate-chunked' strategy (reflected in Cluster 1). Finally, it is important to repeat that the instructions did not include any specific directions on how the targets should be consolidated but merely concerned how the targets would be reported. Therefore, the participants were free to consolidate the two targets in any way they saw fit without breaking any of the instructions, which may have contributed to the unpredictability of strategy choice.

To conclude, the two previous analyses suggest that the instruction prevented participants in the 'combined' condition from adopting the standard 'consolidate-separate' skill, but that it did not prevent the participants in the 'separate' condition from spontaneously also using the same chunked consolidation skill. This again shows the unpredictable nature of strategy manipulations and supports the necessity of the cluster analysis to get an accurate understanding of the data.

**Model fit cluster analysis.**   The unreliable change in strategy following the instruction manipulation prevents direct comparison of the model predictions with participant performance since the model assumes that the manipulation will result in a perfect separation of strategies (i.e., that every participant dutifully uses the instructed strategy). Therefore, we compared the model prediction to the patterns revealed by the cluster analysis. An important side note is that we did not perform any model fitting after data collection, the presented model outcomes are purely predictions made before the experiment was conducted. The performance of the participants in Cluster 1 was compared with the version of the AB model that consolidated the two targets as one chunk and the performance of the participants in Cluster 2 was compared with the version of the AB model that consolidated both targets separately.

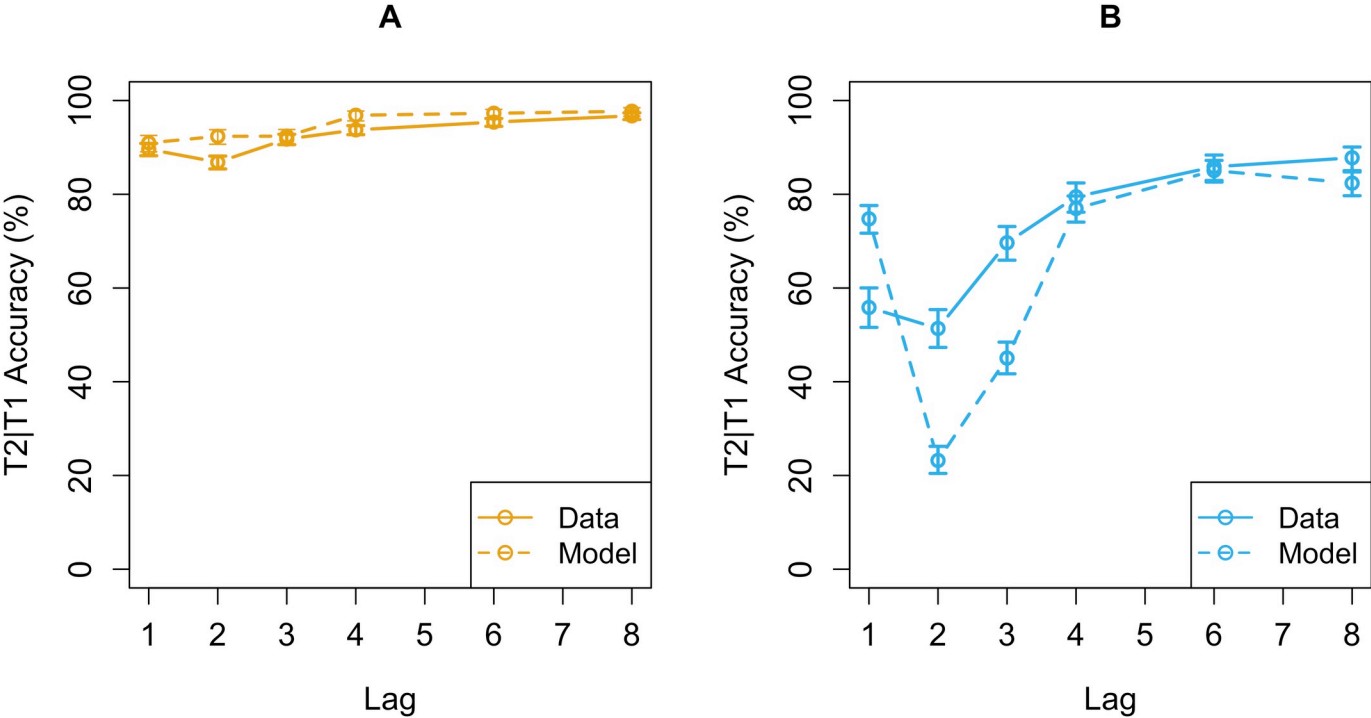

**Fig 9. Model predictions and participant data for Cluster 1 and Cluster 2.** (a) the predictions of the 'consolidate-chunked' version of the AB model (dashed orange line) which lines up well with the performance of the participants from Cluster 1 (solid orange line). (b) the predictions of the 'consolidate-separate' version of the model (dashed blue line) and the performance of the participants in Cluster 2 (solid blue line). Confidence intervals were calculated using the Agresti-Coull method [38].

This strongly improved the fit of the model predictions with the participant data (Fig 9). The performance of the 'consolidate-chunked' version of the AB model lines up well with the performance of the participants in Cluster 1. Both the model and the participants performed very accurately with an accuracy level of around 90% combined with a very small AB (Fig 9A). Additionally, the 'consolidate-separate' version of the model shows the same pattern of performance as the participants in Cluster 2 (Fig 9B). Both the model and the participants show a clear AB with lowest performance at lag 2 and a quick recovery at lag 3.

Although the model predicted the correct direction of the effect, it did strongly overestimate the size of the effect (i.e., the AB magnitude). This might be due to the decreased T1 difficulty in Experiment 2 which might have a mediating effect on the AB [49]. We chose not to account for this in our model because we cannot conclude that T1 difficulty is indeed the reason for the smaller AB and because incorporating a potential mechanism that might be able to explain it would diminish the generalizability of our AB model. A final interesting finding in Cluster 2 was that these participants did not show Lag-1 sparing. Although the cause of this cannot be conclusively determined, it may be due to the quicker target detection in our study relative to other AB studies. This faster target detection may have shifted the AB to the left, leading to worse performance at lag 1 but improved performance at lag 3.

Cluster 1 and Cluster 2 are in line with the model predictions, however our model does not account for the pattern shown by the participants in Cluster 3. The surprisingly low accuracy in this cluster was not predicted by the model and the data and design of the current study cannot give a definite answer as to what caused this performance pattern. However, it might be related to the combined consolidation strategy since it is displayed mainly by participants in the 'combined' condition.

## Discussion

The design of Experiment 1 proved insufficient to change the consolidation strategy employed by the participants during an AB task. Therefore, we conducted Experiment 2 which included targets that promote chunking, in addition to the instruction manipulation already present in Experiment 1 (see Fig 4).

The results of Experiment 2 revealed a significantly reduced AB for the participants who were instructed to remember both targets as a single chunk (i.e., the 'combined' condition) compared to the participants who were instructed to remember both targets separately (i.e., the 'separate' condition). Although this result is in line with the predictions of the model, the modulation of the AB was much weaker than predicted by the AB model and, therefore, this initial analysis does not (fully) support the model predictions.

However, it is likely (as suggested by the results of Experiment 1) that the instruction manipulation has a different effect on individual participants. Therefore, we conducted a cluster analysis on the data in order to gain insight into these individual differences. The cluster analysis revealed three distinct performance patterns: (1) a large group of participants that performed remarkably well on all lags and hardly showed an AB, (2) a group of participants experiencing a mostly regular AB although without Lag-1 sparing and, finally, (3) a group of participants that performed relatively well at Lag 1 but performed very poorly at all subsequent lags.

The first pattern revealed by the cluster analysis is especially intriguing since it shows a very strong modulation of the AB experienced by a large number of participants (52). This opposes the common assumption that the AB reflects a fundamental limitation of cognition and raises the question of how these participants were able to avoid the AB bottleneck. Although it cannot be concluded with certainty, it is unlikely that this strong modulation is caused by the individual ability of the participants, because of the large number of participants in which it was observed. Additionally, it cannot be produced by the stimuli alone because the cluster analysis also revealed a group of participants showing a mostly regular AB. Therefore, it is likely that these participants managed to avoid the AB by using a particular strategy.

We assume that this strategy is the chunked consolidation strategy, because the use of the strategy seems to be directly linked to the targets used in Experiment 2, which were easily combined into a single chunk of information. This characteristic was initially intended to merely support the instruction manipulation, however after data collection it seems likely that the targets also cued the chunked consolidation strategy and that the instruction manipulation was not the only driving factor. Although our data cannot prove without question why the participants in cluster 1 managed to avoid the AB, their performance suggests they adopted the chunked consolidation strategy, which was triggered by a combination of the stimuli and instructions which allowed them to avoid the memory consolidation step thought to underlie the AB.

The performance pattern of the participants in the third cluster was unexpected and the design of the study does not provide a clear answer to why this occurred. However, significantly more participants from the 'combined' condition were assigned to this cluster. This suggests that these participants might have struggled with executing the 'consolidate-chunked' strategy. A crucial difference between the two conditions is that the two targets need to be integrated into one object in the 'combined' condition while this is not necessary in the 'separate' condition. It could be that this integration proved very difficult for these participants and that this led to the low performance at the later lags. The relatively good performance at Lag 1 in this cluster also suggests that the main difficulty with these participants occurs when there is at least one distractor in between the two targets. In accordance with this idea, the participants

from the 'separate' condition in this cluster performed very differently. These participants showed a long and deep blink (as can be seen in Fig 8) which suggests no problems with integration but mainly points to a pronounced AB.

An important limitation of the current study is that the model predictions were only supported in a relatively 'post-hoc' manner. Although we consider the outcome of the cluster analysis to be a better reflection of the data than the partition into the two original conditions, we acknowledge that our initial hypothesis was not confirmed but that we only found support for our model in a different manner than we initially conceived. The post-hoc nature of the cluster analysis increases the uncertainty that the 'chunked' consolidation strategy is responsible for the absence of an AB in Cluster 1. This uncertainty is smaller for Cluster 2 since the 'consolidate-separate' strategy is the strategy that participants are commonly assumed to employ during standard AB tasks [14, 15]. However, the 'consolidate-chunked' strategy is not commonly used to explain performance during AB tasks (although it has been suggested before [12]) and the data of the current study cannot unequivocally prove that this strategy was responsible for the absence of an AB in Cluster 1. An experiment using physiological measures might provide more conclusive answers, pupil dilation data combined with the 'deconvolution' method capable of entangling pupil responses to quickly occurring stimuli might be a good candidate for that [50, 51].

To conclude, our model does not provide a full explanation of the cognitive mechanisms involved in performing the modeled task. This is common in computational models since it is possible to check the model predictions against all aspects of the data and therefore exposes every aspect of the model that is not fully in line with reality. Following the generalizability ambitions of the skill-based approach we decided against adjusting model details after data collection (e.g., its parameters) because it often reduces generalizability to other tasks. Furthermore, refraining from post data collection adjustments also provides an interesting opportunity to learn from the incorrect model assumptions. One crucial aspect the model did not predict correctly was the large individual variation participants within an instruction condition showed. As previously mentioned, we assume this is due to differences in strategy although the data cannot fully exclude other possibilities or confirm that the strategies responsible are the ones we modeled. A second crucial aspect the model did not account for was that the combined instructions would have such a negative effect on so many participants. It seems that these participants struggled with integrating the two targets when there were one or more distractors in between the two targets. These two factors prevented us from analyzing the data in the way we intended and the post hoc nature of the analysis we performed instead reduces the reliability of the conclusions. However, the strong reduction of the AB in such a large group of participants is surprising given the existing literature and the nature of the targets supports the idea that this might be due to a difference in how the targets were consolidated compared to more regular AB tasks.

## General discussion

In a previous paper [3] a model of the AB was created only using skills (pieces of procedural knowledge) that had been created as part of other tasks. This resulted in a more naturalistic and human like model of the AB that succeeded in capturing commonly reported aspects of the data. Additionally, the model offered an explanation for the strong reduction [12, 24, 25] and sometimes complete elimination [26, 29] of the AB under certain experimental conditions. The model accounts for these findings by assuming that participants avoid the AB by using a different consolidation strategy (and skill). Instead of consolidating the two targets separately,

they are consolidated together in a single chunk. This strategy allows participants to bypass the bottleneck of serial consolidation commonly suspected to be responsible for the AB.

In the current paper we attempted to collect empirical evidence supporting our AB model. We conducted a replication study of an experiment which reported that participants did not experience an AB when instructed to remember the two targets as a single syllable [29]. Our replication study failed to show the same effect. This failed replication is valuable to report and shows the unpredictable nature of manipulations aimed at changing participants' strategy. Additionally, the difference in results of two very similar experiments (the original study and our replication) combined with the large individual differences shown in Experiment 2 indicate that the specifics of the design, for example the stimuli or the composition of the sample, may considerably change the outcome of an AB experiment. This strengthens the case for the use of large sample sizes in AB studies and shows the importance of including an investigation into the individual differences present in the sample (e.g., with cluster analysis).

Subsequently, we conducted a second experiment which included targets that facilitated chunking in order to obtain a suitable data set on which we could test our model predictions. The results of this experiment supported these predictions in that a significant AB reduction was found, but not to the extent predicted by the model. We explained this smaller than predicted effect by assuming that the instructions did not have the same effect on every participant. Participants in the 'combined' condition may still have employed the more common 'separate-consolidation' strategy while some participants in the 'separate' condition may have been prompted by the targets to use the 'chunked-consolidation' strategy.

Therefore, we performed a cluster analysis on the data to gain additional insight into these individual performance patterns. This analysis revealed the two performance patterns predicted by our model which are indicative of the two different consolidation strategies. In addition, it revealed a third performance pattern which might be the result of an inadequate execution of the chunked consolidation strategy. In conclusion, the direct manipulation of consolidation strategy only (the instruction manipulation) did not result in full support for our model predictions. However, the experiment included an additional indirect manipulation of consolidation strategy (the nature of the targets), the effect of which was uncovered by the cluster analysis. The combination of the direct and indirect manipulation show that it is possible to circumvent the AB bottleneck when a different strategy than the commonly presumed 'consolidate separate' strategy is adopted.

Most of the components of the AB model were created for a previous paper [3] using a novel modelling approach which involves only (re)using skills taken from other models. This approach is based on the idea that people do not consider a new task in isolation but use previously learned procedural knowledge (skills) to perform the new task. The skill-based approach is one way of bringing that idea into practice and the data collected in Experiment 2 supports its predictions. The model we created improved on two of the core advantages of models created within a cognitive architecture: cognitive plausibility and generalizability. Our model has become more plausible than it would have been without considering skill reuse because the skills we used to model the AB have been verified by using them in other tasks besides the AB task. This reuse also shows that the procedural knowledge (the operators/production rules) used by our AB model is not implausibly (task-) specific. Furthermore, the use of general procedural knowledge has made the findings of our AB model more easily generalizable to other contexts. The fundamental limit to item consolidation can be expected in any situation in which people use the "consolidate-separate" skill (and be avoided by using the "consolidate-chunked" skill).

This requirement that the model can only be created from reusable skills encouraged a simplistic approach to building the model and resulted in a more natural and straightforward

explanation of the AB. In our model, the AB is simply a consequence of normal cognitive functioning and does not require any specific mechanisms primarily aimed at explaining the AB. Additionally, our model offers a new perspective on the AB by showing the importance of strategy during an AB task. This effect of strategy may be the reason why it has been so challenging to arrive at a consensus on the mechanisms behind the AB. Previous models of the AB only propose a single mechanism responsible for the AB and therefore are unable to account for effects caused by differences in strategy. However, our data and model suggest that strategy plays a crucial role and that a singular explanation of the AB might not be sufficient.

Another advantage of the more simplistic modelling approach imposed by the limitations of the skill-based approach is the improved ability to relate the proposed mechanisms to other (general) theories and models. Our AB model is not as extensive and detailed as other AB models nor does it propose a completely original mechanism to be responsible for the AB. However, this mechanism is explicitly defined as part of a general processing step (memory consolidation), which strongly aids the integration of the insights gained from this model with existing theories. As mentioned before, there is no consensus on the mechanism responsible for the AB (e.g., see [52, 53] for alternative accounts), and we do not claim that our model resolves all the differences between the many models of the AB. However, using the skill-based approach, we were able to precisely identify a general cognitive process *potentially* responsible for the AB. This link to more general theories generally lacks in other AB models (which, on the other hand, are often more detailed than our model). However, this link is highly valuable. It strongly improves the generalizability of models and theories which can reduce the division between the different models of the AB and, in general, between the sub fields of cognitive psychology.

To summarize, the main benefit of using the skill-based approach is the improved balance between specificity and generalizability. Models created using the skill-based approach are specific enough to explain a certain phenomenon but, at the same time, are general enough to be easily linked to more general theories and other models.

The skill-based approach in its current form can nevertheless be improved in multiple ways. Firstly, creating models with this approach is more cumbersome and time consuming compared to standard modelling practices. The skills need to be built in a way that suits multiple tasks increasing the difficulty of creating these skills and they need to be verified by creating extra models that use the same skills. Secondly, applying the skill-based approach is complicated by certain assumptions made by many cognitive architectures. For example, the strict rule-based firing of production rules in ACT-R makes it very difficult to develop general production rules that can be used in multiple tasks. PRIMs allows more flexibility because operator selection is partly based on activation, however explicit condition checking is still required for reliable behaviour.

In our future work we will work on improving these aspects of the skill-based approach. However, the two main benefits of this approach can also be achieved by partial implementation of the principles of the skill-based approach. Models can become more flexible and human-like by considering that operators/production-rules are likely to be reused in other contexts when building a model. Additionally, the fact that our AB model could be easily related to other models and general theories is largely due to dividing the cognitive processes involved in the modelled task into general processing steps (i.e., skills).

In conclusion, building cognitive models based on the idea of skill-reuse can create novel insights and presents important improvements to certain aspects of cognitive modelling. In the case of the AB, it has shown that the AB can be a simple consequence of normal cognitive functioning and that it can be avoided using an alternative consolidation strategy. Because the AB model was created with general pieces of procedural knowledge (skills), the model reached

a level of flexibility and robustness which is difficult to achieve without such an approach. Finally, we have shown that the skill-based approach is capable of producing valid models and new predictions.

## Author Contributions

**Conceptualization:** Corné Hoekstra, Sander Martens, Niels A. Taatgen.

**Data curation:** Corné Hoekstra.

**Formal analysis:** Corné Hoekstra.

**Funding acquisition:** Niels A. Taatgen.

**Investigation:** Corné Hoekstra.

**Methodology:** Corné Hoekstra, Sander Martens, Niels A. Taatgen.

**Supervision:** Sander Martens, Niels A. Taatgen.

**Visualization:** Corné Hoekstra.

**Writing – original draft:** Corné Hoekstra.

**Writing – review & editing:** Corné Hoekstra, Sander Martens, Niels A. Taatgen.

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
