## [Decision Letter · Decision Letter 0]

9 Sep 2021

PONE-D-21-21645Testing the Skill-Based Approach: Consolidation strategy impacts Attentional Blink performancePLOS ONE

Dear Dr. Hoekstra,

Thank you for submitting your manuscript to PLOS ONE. After careful consideration, we feel that it has merit but does not fully meet PLOS ONE’s publication criteria as it currently stands. Therefore, we invite you to submit a revised version of the manuscript that addresses the points raised during the review process.

The reviewers found the approach taken in your study to be both theoretically innovative and worthwhile and the experiments appear to be technically sound. Likewise, I also found the approach fascinating and can see its potential value for understanding the attentional blink. However, the reviewers have both raised important concerns, which you will need to address. I won’t reiterate all of them here, but I do want to highlight a common thread that there are inconsistencies between the model predictions and the results and that reconciling these appropriately in your conclusions is an issue that needs further attention. On this point, the reviewers raise related criticisms about the validity of your interpretation of the results, noting that the model is not particularly well supported by the data. While the results of the cluster analysis can be mapped onto the model’s prediction *post hoc*, doing so doesn’t provide confirmation of the validity of the model. I found the results of the cluster analysis interesting but, like Reviewer 1, I’m left wondering how to interpret three apparently distinct clusters of participants when there are only two candidate skills considered in the model. In addition, both reviewers found it somewhat problematic that most of your participants in Experiment 2 had very high accuracy, with the largest cluster showing little evidence of an AB. As they note, this is consistent with your model predictions to a degree, but could be interpreted in many other ways. To summarise these points, then, although the approach of applying PRIMS to the AB is commendable for a number of reasons, this particular study appears to generate more questions than it answers and this should be reflected in your interpretation and conclusion.

In addition to the reviewers’ comments, I couldn’t find a few details that I though would be useful for understanding and/or replicating the results. For instance, I think you should report the font in which the RSVP items are displayed, accuracy for the single-target trials in Experiment 2, and the proportion of trials that were omitted from analysis because the participant chose too many corners (as you have described on pages 21-22). You’ll have to forgive me if these details are buried in the manuscript somewhere but I couldn’t find them and I think they should be in the main text of the article, not just accessible in the data that you have made available.

As the reviewers agree that the study is technically sound, I expect that these concerns can be addressed in a substantial revision of the manuscript, focusing on the interpretation of the results in particular.

We look forward to receiving your revised manuscript.

Kind regards,

Evan James Livesey, Ph.D

Academic Editor

PLOS ONE

Journal Requirements:

Reviewers' comments:

Reviewer's Responses to Questions

**Comments to the Author**

1. Is the manuscript technically sound, and do the data support the conclusions?

Reviewer #1: No

Reviewer #2: Partly

2. Has the statistical analysis been performed appropriately and rigorously? 

Reviewer #1: No

Reviewer #2: Yes

3. Have the authors made all data underlying the findings in their manuscript fully available?

Reviewer #1: Yes

Reviewer #2: Yes

4. Is the manuscript presented in an intelligible fashion and written in standard English?

Reviewer #1: Yes

Reviewer #2: Yes

5. Review Comments to the Author

Reviewer #1: This manuscript tests the ability of the PRIMs cognitive architecture, which is based on a modular set of basic cognitive operations, to explain the attentional blink (AB). The AB is a deficit in the reporting of the second of two consecutive targets that arises when the first target must also be reported. The authors run two experiments in which they use instructional manipulations to encourage participants to treat the two AB targets as distinct or as part of a larger whole. In Experiment 1, this involved asking participants to report two letters or combine those letters to report one syllable. In Experiment 2, this involved asking participants to report corners of a square separately or as a combined shape. In Experiment 1, this manipulation did not affect performance, failing to replicate an earlier study (thus PRIMS could not be tested). In Experiment 2, the manipulation led to some improvement in T2 performance when participants were asked to report the combined shape. Modelling this difference using two PRIMs architectures that used a “consolidate-separate” or “consolidate-chunk” skill (corresponding to the instruction to encode targets separately or as a whole) yielded “model predictions [that] were in the correct direction … [but] the difference between conditions was much smaller than anticipated by the model.” (pg 28).

The authors employ an interesting and flexible approach to explaining and modeling the AB, and the experiments are conducted thoroughly and analyzed appropriately for the most part. My chief concern is that the interpretation of the results seems to move too far past the existing data. Starting with Experiment 1, the failure to replicate the earlier Ferlazzo paper is potentially attributed to a number of factors (lines 401-416), and then the authors conclude that “the failed replication does not categorically reject the original results but rather shows how difficult it is to manipulate participant strategy”. While this is a reasonable conclusion, there does not seem to be anything in the data that supports this conclusion over and above alternatives, including the possibility that the earlier results are not replicable.

In Experiment 2, a similar manipulation to promote chunking of the targets is successful in improving T2 performance at earlier lags, but then leads to worse performance at longer lags. Thus, the model captures the early benefit to some degree but then cannot account for the puzzling deficit in the “consolidate-chunk” condition at later lags. The authors then conduct a cluster analysis and show that there are three clusters of differently performing participants. While I understand that the choice of three clusters is data driven, conceptually this seems odd given that one should expect only two clusters corresponding to the two different instructions. In fact, the three-cluster structure suggests that a significant majority of participants simply showed no AB. Arguably, this makes the outcome of the experiment itself largely uninterpretable and the fit of the model to the data (which was meant to be indicative of an AB) also unreliable. The authors take a different approach suggesting that the large cluster of participants who do not show an AB reflects the fact that many in the “consolidate-separate” condition instead used a “consolidate-chunk” strategy. This is, of course, plausible but there is nothing in the data to support this particular interpretation over any other one.

To conclude then, while I am entirely sympathetic to the authors approach and I think they have conducted their experiments well, I don’t feel that their data clearly supports their model or is evidently consistent with their interpretations.

Reviewer #2: The manuscript reports the results of two experiments, and a computational modelling, aimed at investigating a skill-based model of the Attentional Blink (AB), predicting a reduced or absent AB when individuals use a single-chunk consolidation strategy while performing the AB task. The first experiment is an attempt at replicating the results from a study by Ferlazzo et al. (2007). The second experiment, prompted by the failure at replicating them, is an attempt at devising a better experimental manipulation yielding the expected reduction of the AB. The results, though complex, could be interpreted to support the model.

The study is quite interesting from a theoretical point of view. It's well-designed, and contributes nicely to our knowledge, or lack thereof, on the AB phenomenon.

In my opinion, the main weakness of the manuscript concerns the interpretation of the results from the cluster analysis. Though the analysis is interesting, its results are interpreted as demonstrating the validity of the model the Authors propose. However, the results are just post-hoc (there was no a priori expectation about the results of the cluster analysis) and cannot be used this way. The results of the experiment are indeed in contrast with those expected on the basis of the model (or at least not conclusive), and those from the cluster analysis can only, in my opinion, be used to generate new hypotheses to be investigate in further experiments. This should be made clear in the manuscript by changing the phrasing. For instance:

(from line 782) Specifically, it appears that the participants in the ‘combined’ condition rarely used the standard ‘consolidate-separate’ skill indicated by the very small number of participants showing a regular AB. However, it also shows that quite some participants in the ‘separate’ condition used the ‘consolidate-chunked’ strategy (reflected in Cluster 1).

This sentence is too strong, as an infinite number of other mechanisms could account for the observed pattern of results, and this should be reflected in the sentence.

Minor point:

Lines in Figure 8 are barely discernible, the figure should be redrawn.

In my opinion, the manuscript presents a set of interesting and new results, adding significantly to our understanding of the AB phenomenon, and only requires minor revisions.

6. PLOS authors have the option to publish the peer review history of their article (what does this mean?). If published, this will include your full peer review and any attached files.

Reviewer #1: No

Reviewer #2: No

---

## [Author Response · Author response to Decision Letter 0]

23 Oct 2021

Dear Dr. Livesey,

Thank you for considering our manuscript for publication in PLOS ONE. We appreciate the constructive comments provided by you and the reviewers and they have led to an improved manuscript. In our revision we have focused on modifying the conclusions we draw to reflect more uncertainty and include other potential explanations. Additionally, we have clarified certain ambiguities and added several important details that were missing in the original manuscript. We hope that these modifications will be sufficient to address the concerns raised by you and the reviewers. Below you can find a more detailed response to the points raised in the review. All page and line numbers refer to the revised manuscript with tracked changes.

Editor remarks:

1. You and the reviewers note that the model predictions were only linked to the data post hoc and that there is no clear explanation for Cluster 3. We agree with these concerns and we have added several paragraphs to the manuscript to clarify and reflect these limitations. Regarding the post hoc nature of the model fit we have added lines 908-922 acknowledging the post hoc nature and suggest an experiment which might test the hypotheses generated by the cluster analysis. Additionally, we have added lines 898-907 elaborating on Cluster 3 and present a possible explanation for the performance pattern in this cluster. Furthermore, lines 799-803 also touch on the possibility that participants might spontaneously adopt other cognitive strategies than we and the model assumed.

2. The second point concerns the high T2 accuracy in Cluster 1 and whether this can be taken as evidence for our model. We second that the high performance by such a large group of participants is very surprising and contrasts with the common conception of the AB as a fundamental limit of cognition. In the paragraph on lines 877-881 we present the argument for the case that the different clusters arise from employing different strategies (in essence, Cluster 1 and 2 show qualitatively different performance patterns doing the same task with the same stimuli. Which we think is unlikely to reflect fundamental different levels of individual ability, especially because many previous AB tasks have shown that people are very limited in their ability to identify two rapidly present successive targets). The final two paragraphs of the discussion of Experiment 2 (p. 37-38) are rewritten and reflect the uncertainty surrounding any hard conclusions about the mechanisms responsible for our results. The first paragraph (lines 908-922) discusses the uncertainty surrounding what these strategies exactly entail and that our data cannot unquestionably show that they are the two strategies we predicted. Furthermore, in the final paragraph (line 923-942) the final conclusion has been modified to reflect the uncertainty in crediting any explicit mechanism for the reduced AB. Furthermore, in this conclusion we repeat the two factors underlying our decision to perform a cluster analysis and the post hoc nature of this analysis.

3. Finally, we have added the missing details you mentioned. The font is mentioned on lines 324-325 and 461-462. The single target trials are analyzed on lines 694-697. The proportion of omitted trials are reported on line 535. Furthermore, while analyzing the proportion of omitted trials, we noticed that the number of excluded trials was very high for two participants. These participants have now been removed completely and the outcomes of the statistics we ran and the figures have been adjusted throughout the manuscript. This did not significantly change the results; the main change was that Cluster 3 included two participants less, as both were previously assigned to that cluster.

Reviewer 1

1. The first point raised by the reviewer states that our replication also might have failed because the results are not replicable at all. We agree with this point and have added this possibility to the discussion of Exp. 1 (p. 17). 

2. The second point has been largely discussed in point 2 of the Editor remarks. It relates to the fact that the high performance in Cluster 1 could also have been caused by something else than the ‘consolidate-chunked’ strategy. We agree with this point and we realize that we were too strong in our conclusions in the original manuscript. We have moderated the conclusions in several parts of the revised manuscript to better reflect the uncertainty (e.g., on lines 887-890 and p. 40).

Reviewer 2

1. The first point raised by reviewer 2 concerns the post hoc nature of the model fit. We have discussed this concern in the first point of the Editor remarks and, in addition to the two previously mentioned paragraphs we have slightly changed the phrasing on line 797 as suggested by the reviewer.

2. The second point concerned a request to change Fig. 8 for it to be easier to read by thickening the lines representing the condition averages, which has been done accordingly.

In closing, we would like to thank you for your attention to our manuscript, and we hope that it now meets the publication criteria of PLOS ONE. We look forward to your response.

With kind regards,

Corné Hoekstra

Sander Martens

Niels Taatgen

---

## [Decision Letter · Decision Letter 1]

22 Dec 2021

Testing the Skill-Based Approach: Consolidation strategy impacts Attentional Blink performance

PONE-D-21-21645R1

Dear Dr. Hoekstra,

We’re pleased to inform you that your manuscript has been judged scientifically suitable for publication and will be formally accepted for publication once it meets all outstanding technical requirements.

Kind regards,

Evan James Livesey, Ph.D

Academic Editor

PLOS ONE

Additional Editor Comments (optional):

Please accept my apologies for the long delay in returning this decision (completely my fault, I'm sorry to say). I secured a review from one of the original reviewers, who was satisfied with the revisions that you have made to the manuscript and has recommended accepting this version of the manuscript. I too have read through your emendations carefully. Thank you for adding the additional information that I requested. I note that you have provided a much more measured set of interpretations and conclusions, which I think fit the results appropriately now. In short, I am also happy with this version. The other reviewer did not provide further feedback on your emendations, however it is clear to me that you have addressed their original concerns and since they recommended minor revision in the first place, I will assume they too would be happy with your efforts.

Reviewers' comments:

Reviewer's Responses to Questions

**Comments to the Author**

1. If the authors have adequately addressed your comments raised in a previous round of review and you feel that this manuscript is now acceptable for publication, you may indicate that here to bypass the “Comments to the Author” section, enter your conflict of interest statement in the “Confidential to Editor” section, and submit your "Accept" recommendation.

Reviewer #1: All comments have been addressed

2. Is the manuscript technically sound, and do the data support the conclusions?

Reviewer #1: Yes

3. Has the statistical analysis been performed appropriately and rigorously? 

Reviewer #1: Yes

4. Have the authors made all data underlying the findings in their manuscript fully available?

Reviewer #1: Yes

5. Is the manuscript presented in an intelligible fashion and written in standard English?

Reviewer #1: Yes

6. Review Comments to the Author

Reviewer #1: (No Response)

7. PLOS authors have the option to publish the peer review history of their article (what does this mean?). If published, this will include your full peer review and any attached files.

Reviewer #1: No

---

## [Editor Report · Acceptance letter]

10 Jan 2022

PONE-D-21-21645R1 

Testing the Skill-Based Approach: Consolidation strategy impacts Attentional Blink performance 

Dear Dr. Hoekstra:

I'm pleased to inform you that your manuscript has been deemed suitable for publication in PLOS ONE. Congratulations! Your manuscript is now with our production department. 

Kind regards, 

on behalf of

Dr. Evan James Livesey 

Academic Editor

PLOS ONE